# Identification of Xanthine Oxidase Inhibitors from Celery Seeds Using Affinity Ultrafiltration–Liquid Chromatography–Mass Spectrometry

**DOI:** 10.3390/molecules28166048

**Published:** 2023-08-14

**Authors:** Xiaona Gan, Bo Peng, Liang Chen, Yanjun Jiang, Tingzhao Li, Bo Li, Xiaodong Liu

**Affiliations:** 1Nutrilite Health Institute, Amway (China) R&D Center, Shanghai 201203, China; breeze56837@126.com (X.G.); shawn.peng@amway.com (B.P.); clark.chen@amway.com (L.C.); teric.li@amway.com (T.L.); 2Department of Anaesthesia and Intensive Care, The Chinese University of Hong Kong, Hong Kong SAR, China; jiangyanjun111@outlook.com; 3Peter Hung Pain Research Institute, The Chinese University of Hong Kong, Hong Kong SAR, China

**Keywords:** celery seeds, UF–LC–MS, xanthine oxidase inhibitors

## Abstract

Celery seeds have been used as an effective dietary supplement to manage hyperuricemia and diminish gout recurrence. Xanthine oxidase (XOD), the critical enzyme responsible for uric acid production, represents the most promising target for anti-hyperuricemia in clinical practice. In this study, we aimed to establish a method based on affinity ultrafiltration–liquid chromatography–mass spectrometry (UF–LC–MS) to directly and rapidly identify the bioactive compounds contributing to the XOD-inhibitory effects of celery seed crude extracts. Chemical profiling of celery seed extracts was performed using UPLC-TOF/MS. The structure was elucidated by matching the multistage fragment ion data to the database and publications of high-resolution natural product mass spectrometry. Thirty-two compounds, including fourteen flavonoids and six phenylpeptides, were identified from celery seed extracts. UF–LC–MS showed that luteolin-7-*O*-apinosyl glucoside, luteolin-7-*O*-glucoside, luteolin-7-*O*-malonyl apinoside, luteolin-7-*O*-6′-malonyl glucoside, luteolin, apigenin, and chrysoeriol were potential binding compounds of XOD. A further enzyme activity assay demonstrated that celery seed extract (IC_50_ = 1.98 mg/mL), luteolin-7-*O*-apinosyl glucoside (IC_50_ = 3140.51 μmol/L), luteolin-7-*O*-glucoside (IC_50_ = 975.83 μmol/L), luteolin-7-*O*-6′-malonyl glucoside (IC_50_ = 2018.37 μmol/L), luteolin (IC_50_ = 69.23 μmol/L), apigenin (IC_50_ = 92.56 μmol/L), and chrysoeriol (IC_50_ = 40.52 μmol/L) could dose-dependently inhibit XOD activities. This study highlighted UF–LC–MS as a useful platform for screening novel XOD inhibitors and revealed the chemical basis of celery seed as an anti-gout dietary supplement.

## 1. Introduction

Gout is a common type of arthritis with increasing prevalence in China. Gout stones formed by monosodium urate (MSU) deposit in the joints, especially the toes, causing inflammatory arthritis and severe pain in the affected joints [1]. The cumulative incidence of gout attacks is positively correlated with serum uric acid (sUA) levels in patients. Beyond the saturation point of MSU (68 mg/L), new MSU crystals can form in joints. In contrast, a low sUA (<60 mg/L) leads to the dissolution of existing crystals, reducing the risk of gout recurrence [2]. In this context, medicines that control sUA have been widely used in clinical practice to prevent gout recurrence [3]. Xanthine oxidase (XOD) is the rate-limiting enzyme in uric acid production from purine metabolism [4]. XOD inhibitors such as allopurinol and febuxostat efficiently reduce sUA and are therefore often used to treat gout. Currently, XOD remains the most promising therapeutic target for the development of novel anti-gout drugs.

Natural products are recognized as important sources of bioactive compounds. Celery seed (*Apium graveolens* L.) is rich in various phytochemicals such as flavones [5], coumarin [6], volatile oils [7], and fatty acids [8], making it a popular dietary supplement with antioxidant, anti-inflammatory, hypoglycemic, hypolipidemic, and antihypertensive effects [8,9]. Recently, the anti-gout effects of celery seed have received increasing attention due to its role in XOD inhibition and sUA lowering [10,11]. Meanwhile, the chemical profiles of celery seed extracts were predicted to contain potential XOD inhibitors in a neural network drug-target interaction model, supporting the anti-gout effects of celery seed [12]. The conventional methods for isolating XOD inhibitors from celery seed extracts, which require multistep procedures including compound separation and biological activity assays, are costly and time-consuming. On the other hand, affinity ultrafiltration–liquid chromatography–mass spectrometry (UF–LC–MS), simultaneously exploiting protein–ligand interaction and chemical analysis methods, could be a straightforward approach to isolate novel enzyme modulators from chemistry libraries or plant extracts [13]. UF–LC–MS is now widely used to screen drugs for diabetics [14,15], hyperlipidemia [16], and inflammatory disorders [17]. In this study, we applied UPLC-TOF/MS to profile celery seed chemical compositions and developed a UF–LC–MS-based method to screen the compounds contributing to celery seed-mediated XOD inhibition.

## 2. Results

### 2.1. Chemical Profiles of Celery Seed Extracts

Celery seed extracts were analyzed using UPLC-Q-TOF/MS. The multi-level structural characterization of reconstituted extracts was matched to the database and publications on high-resolution natural product mass spectrometry. A total of 32 compounds were identified, including 14 flavonoids, 6 phenylpeptides, and 3 quinic acids. The total ion current chromatogram (TICC) of the extracts in positive and negative models was shown in Figure 1. The 32 peaks were elucidated and are reported in Table 1. As examples, the MS analyses of luteolin 7-O-malonyl-apiosylglucoside and sedenenolide are demonstrated in Figure 2.

### 2.2. Identification of XOD-Binding Compounds from Celery Seed Extracts

After incubation with febuxostat- (masked) and vehicle- (unmasked) treated XOD, the chromatogram of two crude extract groups demonstrated a series of peaks with different areas. Based on the variation in the chromatogram with and without febuxostat binding in the XOD, the binding degree was used to evaluate the febuxostat competitive capacity of compounds in celery seeds. The binding degree (BD) was calculated as follows:$$BD(\%)=\frac{Af-Ano\_f}{Af}\times 100\%$$

*Af* and *Ano_f* represent the peak area of compounds in the chromatogram with and without preincubation with febuxostat, respectively. The results showed that the binding degree of compound 15 (99.0%), 21 (97.8%), 25 (96.7%), 10 (95.9%), 24 (95.5%), 14 (93.6%), and 9 (89.1%) was greater than 80%, followed by compounds 19 (63.2%), 20 (52.0%), and 11 (50.5%). Specifically, luteolin-7-*O*-apinosyl glucoside (**9**), luteolin-7-*O*-glucoside (**10**), luteolin-7-*O*-malonyl apinoside (**14**), luteolin-7-*O*-6′-malonyl glucoside (**15**), luteolin (**21**), apigenin (**24**), and chrysoeriol (**25**) are potential ligands of febuxostat binding domain in the XOD (Figure 3). The extracted ion current chromatogram and structural formula of the candidate ligand from celery seeds are shown in Figure 3 and Figure 4.

### 2.3. Validation of Inhibitory Activities

The inhibitory effects of crude extracts and six binding compounds on XOD enzymatic activities were assessed as described in the methods. As shown in Figure 5, the extracts (250–8000 μg/mL), luteolin-7-*O*-apinosyl glucoside (448–4480 μmol/L), luteolin-7-*O*-glucoside (156–4480 μmol/L), luteolin-7-*O*-6′-malonyl glucoside (156–4480 μmol/L), luteolin (3–729 μmol/L), apigenin (3–1458 μmol/L), and chrysoeriol (3–1458 μmol/L) dose-dependently inhibited XOD activities. The IC_50_ of celery seed extract is 1.98 mg/mL, and inhibitory activities of compounds were ranked from largest to smallest: chrysoeriol (IC_50_ = 40.52 μmol/L), luteolin (IC_50_ = 69.23 μmol/L), apigenin (IC_50_ = 92.96 μmol/L), luteolin-7-*O*-glucoside (IC_50_ = 975.83 μmol/L), luteolin-7-*O*-6′-malonyl glucoside (IC_50_ = 2018.37 μmol/L), and luteolin-7-*O*-apinosyl glucoside (IC_50_ = 3140.51 μmol/L).

### 2.4. The Study of Molecular Recognition

Molecule docking is a convenient and effective way to explore the interaction of small molecules with targets. Here, we used Vina 1.1.2 to model and calculate the binding capacity between luteolin-7-*O*-apinosyl glucoside (9), luteolin-7-*O*-glucoside (10), luteolin-7-*O*-malonyl apinoside (14), luteolin-7-*O*-6′-malonyl glucoside (15), luteolin (21), apigenin (24), or chrysoeriol (25) and xanthine dehydrogenase. The in silico binding energies were −4.9, −9.4, −4.1, −7.8, −10.8, −10.4, and −9.8 kcal/mol, respectively. The results of the virtual screening showed that the scores of flavonoid aglycones were higher than flavonoid glycosides, which was consistent with the results of in vitro experiments. In this study, the flavonoid glycoside (10) and aglycone (25) with the best activity were selected to reveal the interaction mechanism between compounds and xanthine oxidase.

In Figure 6, 3D and 2D diagrams of the binding mode of luteolin-7-*O*-glucoside (10)/xanthine dehydrogenase(A-B) and chrysoeriol (25)/xanthine dehydrogenase(C-D) complex are shown. These complexes maintain mutual attraction mainly through hydrogen bonds, pi-pi stacking (presumptive noncovalent pi interactions between the pi bonds of aromatic rings), and hydrophobic interaction. For luteolin-7-*O*-glucoside (10)/xanthine dehydrogenase complex (A-B), Arginine (ARG)-880, Threonine (THR)-1010, Glutamic acid (GLU)-897, and HIE-875 (HIE refers to histidine with hydrogen on the epsilon nitrogen) frequently interact with luteolin-7-*O*-glucoside (10) to form hydrogen bonds. In addition, luteolin-7-*O*-glucoside (10) formed pi-pi stack interactions with residues Phenylalanine (PHE)-914 and PHE-1009. For chrysoeriol (25)/xanthine dehydrogenase complex (C-D), small molecules form hydrogen bonds with ARG-880, THR-1010, Valine (VAL)-1011, and GLU-1261 on protein. Moreover, we can also observe that luteolin-7-*O*-glucoside (10) forms pi-pi stack interactions with residues PHE-914 and PHE-1009. Both compounds have hydrophobic interactions with residues such as PHE-1009, Alanine (ALA)-1079, ALA-1078, PHE-914, and PRO-1076 adjacent to MOS.

## 3. Discussion

### 3.1. Identification of Chemical Constituents in Celery Seed Extracts

The chemical composition analysis results of celery seeds show that flavonoids, including luteolin, apigenin, and chrysoeriol and its derivatives, are the main constituents in celery seed ethanol extract. This result is consistent with the results obtained through traditional separation and purification in previous reports [30]. Flavonoid glycoside defragmentation is generally cleaved through glycoside. For example, under the negative ionization mode for luteolin 7-*O*-malonyl-apiosylglucoside, the [M-H]^−^
*m*/*z* 667.1488 was broken down to *m*/*z* 535.1115 fragment ion with the loss of apiosyl, followed by production of *m*/*z* 287.0558 fragment ion after malonyl glucoside breakdown. The phenylpeptides in celery seeds, which contain ester groups and often carry hydroxyl or butyl groups, are prone to generating fragmented ion peaks that remove C_4_H_8_, C_3_H_6_, CO, and H_2_O, such as sedenolide, with a molecular ion of *m*/*z* 193.122 2 [M+H]^+^ and characteristic fragment ions of 175.112 6 [M+H-H2O]^+^; 147.117 5 [M+H-H_2_O-CO]^+^; 137.060 3 [M+H-C_4_H_8_]^+^; 105.069 6 [M+H-H_2_O-CO-C_3_H_6_]^+^; 91.054 0 [M+H-C_4_H_8_-HCOOH]^+^; and 77.038 3 [M+H-C_4_H_8_-HCOOH-CH_2_]^+^.

### 3.2. Identification of XOD-Binding Compounds from Celery Seed Extracts

Screening with the UF–LC–MS method, luteolin-7-*O*-apinosyl glucoside (9), luteolin-7-*O*-glucoside (10), luteolin-7-*O*-malonyl apinoside (14), luteolin-7-*O*-6′-malonyl glucoside (15), luteolin (21), apigenin (24), and chrysoeriol (25) are identified as potential ligands of febuxostat binding domain in the XOD, which indicated that flavone aglycones and flavonoids derived from luteolin were potential XOD inhibitors in celery seeds. As another common chemical class in celery seeds [11], phenylpeptides have rarely been reported in previous studies as having xanthine oxidase-inhibitory activity. In the present UF–LC–MS study, we did not identify any significant difference in the peak area corresponding to phenylpeptides between XOD with and without febuxostat. This result suggests that phenylpeptides in celery seeds exert weak or no competitive blocking activity on XOD.

### 3.3. Validation of Inhibitory Activities

Flavonoids are common polyphenols in plants and have a wide range of pharmacological effects [31]. Studies have shown that the activity of flavonoids is closely related to their chemical structure [32,33]. In this study, the structure–activity analysis of the hitting compounds showed that the inhibitory effects of compounds on XOD appear to be positively associated with the number of hydroxyl groups on ring B. On the other hand, higher glycosylation in flavonoids represents weaker XOD-inhibitory activities. For example, luteolin-7-*O*-apinosyl glucoside (9), luteolin-7-*O*-glucoside, and luteolin-7-*O*-6′-malonyl glucoside (15) are derivatives of luteolin (21), differing only in the number of glycosides. The enzymatic assay demonstrated that luteolin (21) has lower IC_50_ values than its derivatives.

### 3.4. The Study of Molecular Recognition

By predicting the binding strength and affinity between compounds and target proteins, molecular docking can quickly identify active ingredients in plants, and has unique advantages in the virtual screening of active ingredients based on targets. The lower docking affinity reflects the stronger binding ability between the compound and its targets. Shown as 2.4, the binding energies of luteolin-7-*O*-malonyl apinoside (−4.1 kcal/mol) > luteolin-7-*O*-apinosyl glucoside (−4.9 kcal/mol) > luteolin-7-*O*-6′- malonyl glucoside (−7.8 kcal/mol) > luteolin-7-*O*-glucoside (−9.4 kcal/mol) > chrysoeriol (−9.8 kcal/mol) > apigenin (−10.4 kcal/mol) > luteolin (−10.8 kcal/mol). The results indicated that the scores of flavonoid aglycones were higher than flavonoid glycosides, which further supported the findings of in vitro experiments. UF–LC–MS assay combined with molecular docking technology can realize rapid screening of candidates for some drugs.

### 3.5. UF–LC–MS Assay for Screening Novel XOD Inhibitors

The traditional method for screening xanthine oxidase inhibitors from plant extracts involves multiple laborious steps, including extraction, purification, and determination of the compound library and activity detection [30,34]. Compared to traditional procedures, UF–LC–MS combines affinity capture of active molecules against protein targets of interest with ultrafiltration, liquid chromatography, and high-resolution mass spectrometry, which can simultaneously screen and identify bioactive components from complex mixtures, greatly improving efficiency, and reducing sample and reagent consumption [35,36].

Many previous affinity UF–LC–MS studies used mass spectrometry to identify the compounds in the extracts and determined the candidates by comparing the peak area of each compound in the liquid chromatogram [37,38]. However, co-elution and peak overlapping often occur in the liquid chromatogram [39], and some compounds, such as saponins, alkaloids, and amines, have poor UV absorption, resulting in missed screening. The extracted ion current chromatogram used in this study selects specific ion fragments for peak area comparison, effectively avoiding the pitfalls of conventional methods and increasing the hitting rate.

Febuxostat is one of the most commonly prescribed medications to treat gout and hyperuricemia due to its potent inhibitory effect on XOD. Febuxostat is considered superior to allopurinol at standard doses, particularly in patients who are intolerant to allopurinol [40]. Mechanistically, febuxostat non-competitively binds to the molybdenum pterin site of XOD, thereby inhibiting the oxidase activity and preventing the production of uric acid. In this regard, compounds that can competitively displace febuxostat binding in the XOD are candidates for anti-hyperuricemia drugs. The present affinity UF–LC–MS assay compares the binding profiles of XOD with and without febuxostat. The differential compounds between two profiles are therefore considered as candidates sharing a similar binding domain of XOD with febuxostat. A subsequent enzymatic assay confirmed that the potential molybdenum–pterin binding compounds inhibited XOD activity. Taken together, we concluded that UF–LC–MS assay is suitable for the rapid identification of novel XOD inhibitors from crude plant extracts.

## 4. Materials and Methods

### 4.1. Materials and Reagents

Celery seeds were provided by Amway (China) R&D Center (Shanghai, China). The plant origin was identified as *Apium graveolens* L. by lead scientist Dr. Liang Chen of Amway (China) R&D Center. Acetic acid (mass spectrometry grade) was obtained from Thermo Fisher Scientific Inc. (Waltham, MA, USA). Ethanol (AR) was purchased from Titan Technology (Shanghai, China). Methanol and acetonitrile (mass spectrometry grade) were products of Merk (Shanghai, China). Xanthine oxidase (Lot# SLCB1289) was ordered from Sigma-Aldrich Co., Ltd. (St. Louis, MO, USA). Xanthine (Lot# P1356363) and febuxostat (Cat#101569A, Lot# P1125846) were obtained from Adamas (Shanghai, China). Luteolin-7-*O*-glucoside (Lot#6567), Luteolin (Lot#9796), Apigenin (Lot#8758), Chrysoeriol (Lot#10595), Luteolin-7-*O*-malonyl apinoside (Lot#10613), and Luteolin-7-*O*-6′-malonyl glucoside (Lot#10590) were purchased from Nature Standard (Shanghai, China). All chemicals for XO activity assays were AR grade.

### 4.2. Instruments

Agilent 1290 Infinity II ultra-high performance liquid chromatography was obtained from Agilent Technologies (Beijing, China). Waters ACQUITY HSS T3 column (100 × 2.1 mm) was ordered from Waters Corporation (Milford, MA, USA). The Triple TOF 4600 system was purchased from SCIEX (Framingham, MA, USA). The Precision^TM^ Circulating Water Bath was a product of ThermoFisher Scientific (Shanghai, China). Microcon-10kDa centrifugal filter units with Ultracel-10 membrane were ordered from Millipore Co., Ltd. (Bedford, MA, USA). An H1650 centrifuge was obtained from CENCE (Changsha, Hunan, China). A TF-SFD-5PLC freeze dryer was ordered from Shanghai Tianfeng Industrial Co., Ltd. (Shanghai, China).

### 4.3. Chromatographic Analysis

Reconstituted extracts were separated on a Waters ACQUITY HSS T3 column (100 × 2.1 mm, 1.8 μm). The column oven temperature was maintained at 30 °C. The flow rate was 0.2 mL/min. The mobile phase consisted of a combination of A (0.2% acetic acid in water) and B (acetonitrile). The elution gradient was used as follows: 92% A for 0–3 min, 92%-70% A for 3–23 min, 70%-10% A for 23–30 min, and 10–92% A for 30–35 min.

### 4.4. Mass Spectrometry Conditions

AB Sciex Triple TOF^®^ 4600 (Framingham, MA, USA) high-resolution mass spectrometry connected to the Agilent 1290 UHPLC instrument through the ESI interface was used to perform MS and MSn analyses. The mass spectrometer was operated in both positive and negative ion modes. The MS parameter values were set as follows: TOF mass range, 50–1700; ion source gas 1 (psi), 50; ion source gas 2 (psi), 50; curtain gas (psi), 35; ion spray voltage floating (V), −4500/5000; ion source temperature (°C), 500; declustering potential (V), 100; collision energy (eV), 10. The MS/MS parameter values were used as follows: MS/MS mass range, 50–1250; declustering potential (V), 100; collision energy (eV), ±40; collision energy spread (eV), 20; ion release delay (ms), 30; ion release width (ms), 15.

### 4.5. Preparation of Celery Seed Extract

Dried celery seeds were extracted twice at 85 °C in a water bath, the first time with 8 times the amount of 70% (*v*/*v*) ethanol for 1.5 h, and the second time with 6 times the amount of 70% (*v*/*v*) ethanol for 1 h. The ethanol extracts were combined and then concentrated using rotary vaporization at 60 °C under reduced pressure. The extract powder was obtained by drying the concentrated solution using a freeze dryer and stored in a refrigerator at −20 °C until use. For analysis in LC–MS, the extract powder was ultrasonically dissolved in 70% (*v*/*v*) methanol to make a 2 mg/mL solution, followed by filtration before use.

### 4.6. XOD Activity Assay

The inhibitory effects of compounds were determined in the XOD activity assay according to a previous report [41] with minor modifications. Celery seed extracts or the “hit” compounds from the UF–LC–MS screening assay were dissolved in PBS and prepared for a series of test solutions using 2-fold serial dilutions. Test solutions (50 μL) were mixed with XOD (50 μL at 0.02 U/mL) and incubated at 37 °C for 5 min. This was followed by the addition of XOD substrate xanthine (150 μL at 0.48 mM) and further incubation at 37 °C for 30 min. The optical density (OD) of each tube was then measured at 290 nm. Percent XOD inhibition was calculated using the following formula: $\left(\%\right)=\left(1-\frac{A1-A2}{A3-A4}\right)\times 100$ where A1 was the absorbance of the production with the sample and XOD, A2 was the absorbance of the production with the sample, A3 was the absorbance of the production with PBS and XOD, and A4 was the absorbance of the production with PBS.

### 4.7. Screening for XOD Inhibitors Using UF–LC–MS

Identification of potential XOD inhibitors from celery seed extract was performed using UF–LC–MS according to a previous study with modification [42]. To prepare the XOD interaction solution, 250 μL of XOD (0.5 U/mL) was incubated with 50 μL of febuxostat (the XOD inhibitor, 500 μg/mL in 10% DMSO-PBS, masked group) to mask the druggable binding domain of XOD or 10% DMSO-PBS (vehicle control, unmasked group) at 37 °C for 45 min. The extract (50 μL at 2 mg/mL in methanol) was then added to the interaction solution for a further 45 min. After incubation, the mixtures were transferred to a 10 kDa molecule weight cut-off ultrafiltration filter and centrifuged at 14,500 rpm for 20 min. The membrane was washed 3 times with 450 μL ddH_2_O to remove the unbound molecules. The binding molecules, including febuxostat, selectively and unselectively binding compounds, were dissociated by treatment with 200 μL of 70% methanol for 10 min under ultrasonication. The elution solution was then ultrafiltered in the new filter units as described above. The final products in the bottom of the filter units were analyzed using a HPLC–MS system.

### 4.8. Molecular Docking

AutoDock Vina 1.1.2 program [43] was used to model the interaction between the active compounds and the xanthine dehydrogenase protein. The crystal structure of xanthine dehydrogenase was downloaded from the Protein Data Bank database (PDB ID: 1N5X) and pre-processed with PyMol 2.5.2 [44] to remove water molecules and ligands and add hydrogen atoms. The Mo-pt, FE/SI, and FAD structures in the crystal structure were retained. Then, for them, a grid box size of 25 × 25 × 25 Å3 containing the entire binding site was set. Before molecular docking, all proteins and compounds were converted to PDBQT format using AutoDock Tools 1.5.6 program [45], and the charge information of Mo ions was manually set. Docking was performed with default parameters, and the pose with the lowest score was extracted and visualized using Maestro (Schrödinger, New York, NY, USA).

### 4.9. Statistical Analysis

The data were acquired using Analyst TF (version 1.7.1., AB SCIEX, Framingham, MA, USA) and processed with PeakView (version 1.2.0.3, AB SCIEX, Framingham, MA, USA). The GraphPad Prism (version 8.2.1, GraphPad Software Inc., San Diego, CA, USA) was used for statistical analysis and plotting.

## 5. Conclusions

Celery seed is a common spice rich in volatile oil and flavonoids. In this study, a comprehensive LC–MS analysis of the chemical profile confirmed the presence of various flavonoids, phenylpeptides, and quinic acids in celery seed. Using UF–LC–MS techniques, we successfully established a screening assay to identify XOD inhibitors from celery seed extracts. We found that flavonoids represent the major class of compounds in celery seed with potential inhibitory effects on XOD and identified Luteolin-7-*O*-6‘- malonyl glucoside as a new XOD inhibitor for the first time. Furthermore, the XOD activity assay confirmed that these binding compounds inhibited XOD activities in a dose-dependent manner. Our study revealed the chemical basis of celery seed-mediated anti-gout and anti-hyperuricemia effects and provided potential XOD inhibitors for the prevention of recurrent gout.

## Figures and Tables

**Figure 1 molecules-28-06048-f001:**
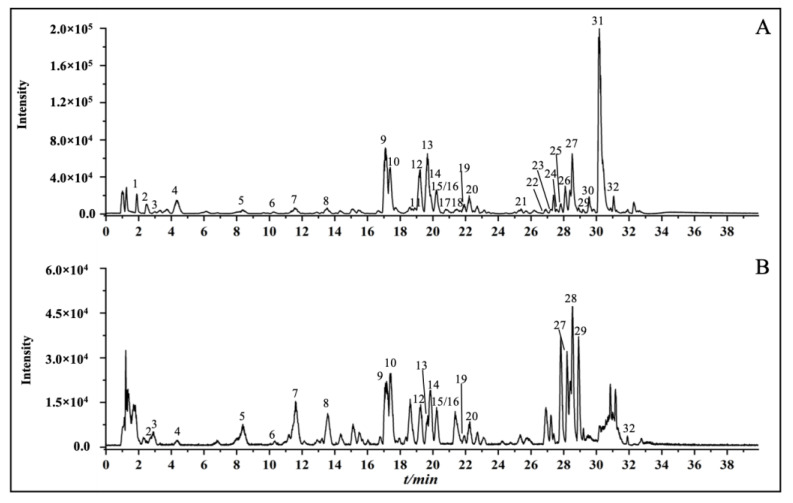
The total ion current chromatogram of celery seed extracts. (**A**) The positive ion mode. (**B**) The negative ion mode.

**Figure 2 molecules-28-06048-f002:**
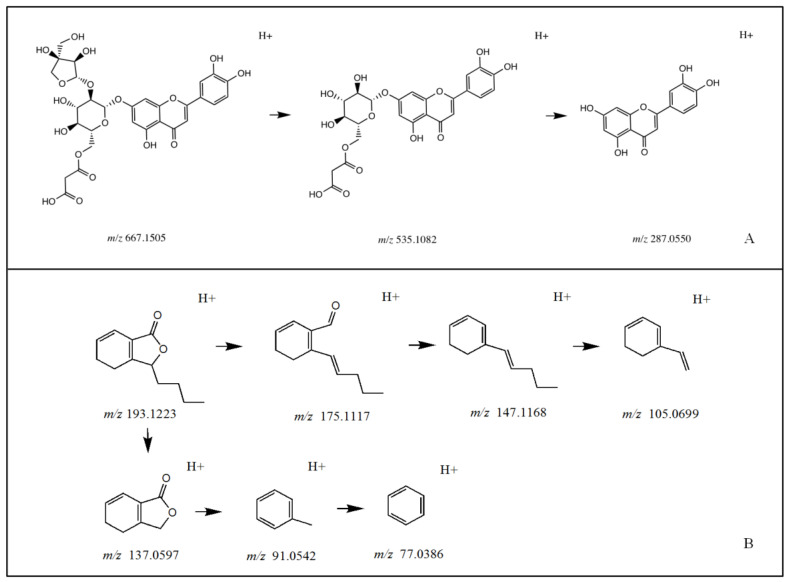
Examples of structure elucidation. (**A**) Luteolin 7-O-malonyl-apiosylglucoside. (**B**) Sedenenolide.

**Figure 3 molecules-28-06048-f003:**
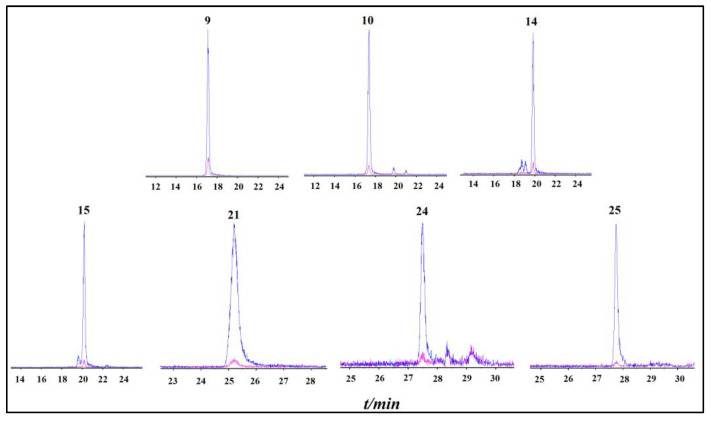
Extracted ion current chromatogram of XOD ligands from celery seeds. Red and blue peaks denote the febuxostat- and DMSO-treated groups, respectively. Numbers refer to peak numbers in Table 1.

**Figure 4 molecules-28-06048-f004:**
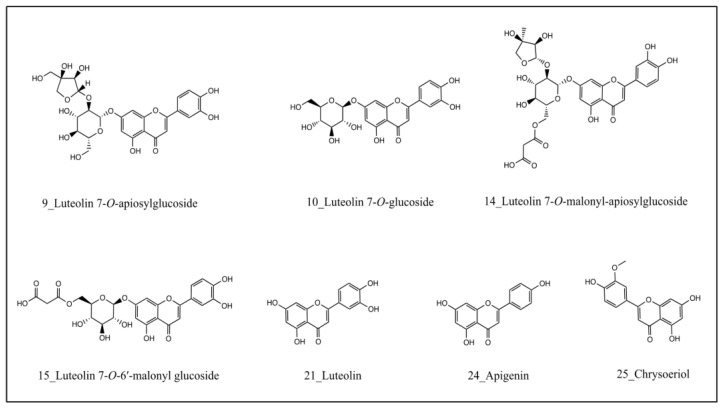
The structural formula of active compounds in celery seeds.

**Figure 5 molecules-28-06048-f005:**
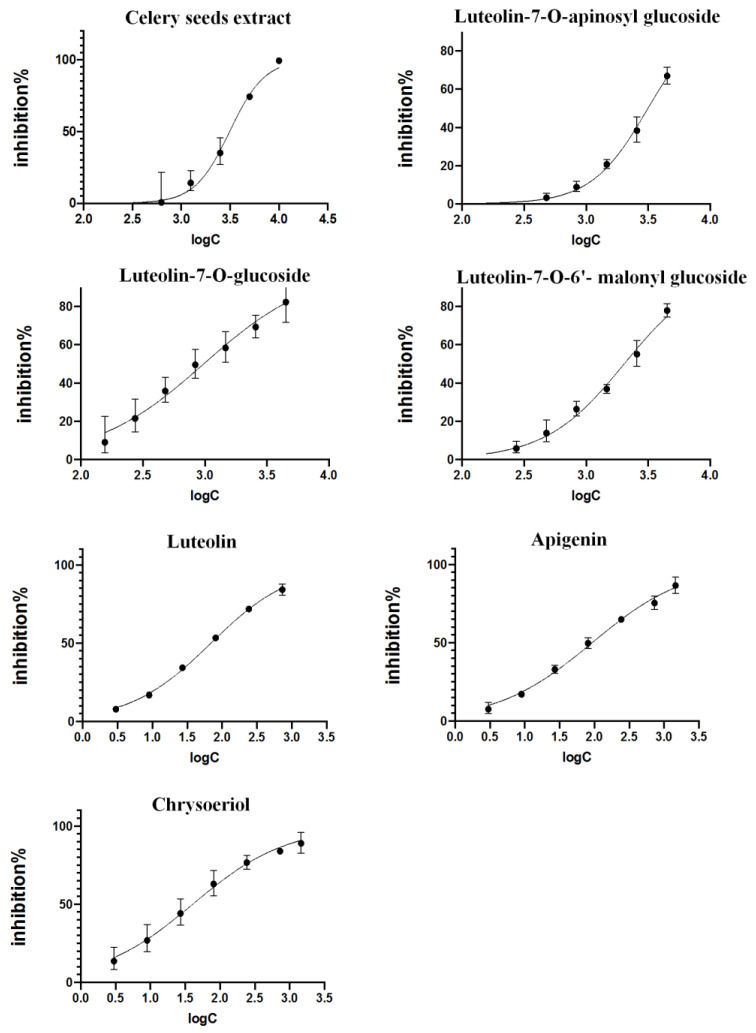
Inhibitory effect of celery seed extract and active compounds on xanthine oxidase.

**Figure 6 molecules-28-06048-f006:**
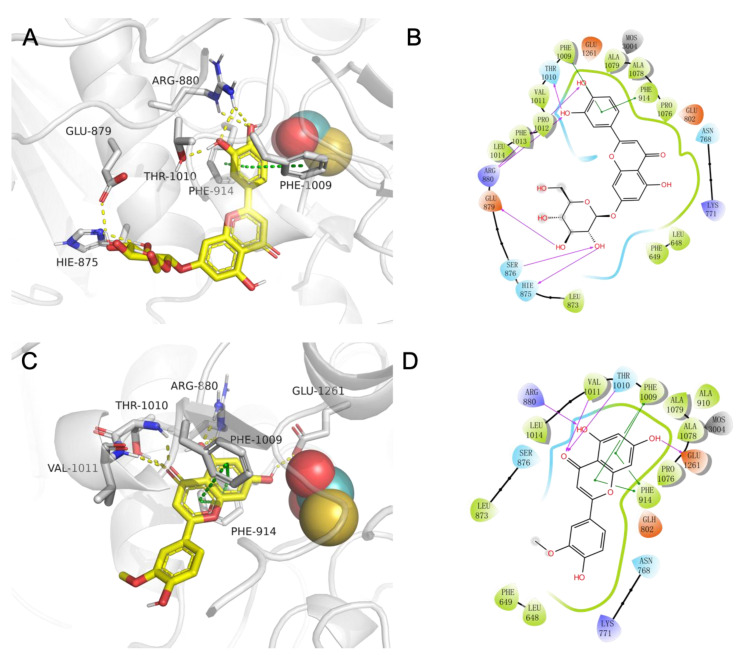
The binding mode of ligand–receptor predicted by AutoDock Vina 1.1.2. The 3D interactions of compound 10/xanthine dehydrogenase (**A**) and compound 25/xanthine dehydrogenase (**C**). The protein was rendered in cartoon and colored in white, while the compound was rendered in stick and colored in yellow. The 2D interactions of compound 10/xanthine dehydrogenase (**B**) and compound 25/xanthine dehydrogenase (**D**). The red dotted line/arrows represent hydrogen bonding; the green dotted line/line represent pi-pi stack interactions; green residues represent hydrophobic amino acids; green edge lines represent hydrophobic contacts; and blue edge lines represent electropositive contacts.

**Table 1 molecules-28-06048-t001:** Characterization of chemical constituents of celery seed extract.

NO	Rt (min)	Identification	*MS*^+^ (*m*/*z*)	ppm	Element Composition	MS2 Fragment Ion (*m*/*z*)	References
1	1.89	Adenosine	268.1042	0.6	C_10_H_13_N_5_O_4_	136.0620; 119.0350	*
2	2.50	Fructose-phenylalanine	328.1390	−0.2	C_15_H_21_NO_7_	310.1326; 292.1213; 254.1228; 246.1097; 166.0861; 132.0792	[18]
3	2.99	Pantothenic acid	220.1176	−1.6	C_9_H_17_NO_5_	220.1164; 202.1063; 184.0972; 128.0257; 90.0550	*
4	4.38	Tryptophan	205.0972	−0.8	C_11_H_12_N_2_O_2_	188.0696; 146.0593; 118.0645; 115.0539	[19] *
5	8.35	Chlorogenic acid	355.1020	−1.0	C_16_H_18_O_9_	NA	[20] *
6	10.62	p-Hydroxybenzaldehyde	123.0441	0.4	C_7_H_6_O_2_	123.0465; 95.0482; 77.0383	[21] *
7	11.54	Coumaroylquinic acid	339.1075	0.2	C_16_H_18_O_8_	147.0436; 119.0481; 91.0532	[20]
8	13.54	Coumaroylquinic acid or isomer	339.1076	0.5	C_16_H_18_O_8_	147.0440; 119.0489; 91.0541	[20]
9	17.11	Luteolin 7-*O*-apiosylglucoside	581.1502	0.2	C_26_H_28_O_15_	287.0554	[20]
10	17.38	Luteolin 7-*O*-glucoside	449.1075	−0.8	C_21_H_20_O_11_	287.0543	[20] *
11	18.91	Senkyunolide J	227.1274	−1.7	C_12_H_18_O_4_	209.1164; 191.1134; 153.0542	[19,22]
12	19.20	Apiin	565.1553	0.2	C_26_H_28_O_14_	271.0609	[20] *
13	19.65	Chrysoeriol 7-*O*-apiosylglucoside	595.1652	−0.9	C_27_H_30_O_15_	463.1226; 301.0701; 286.0463; 258.0526	[20]
14	19.85	Luteolin 7-*O*-malonyl-apiosylglucoside	667.1488	−2.5	C_29_H_30_O_18_	535.1115; 287.0558;	[20]
15	20.22	Luteolin 7-*O*-6′-malonyl glucoside	535.1082	−0.1	C_24_H_22_O_14_	535.1062; 449.1039; 287.0529	[20]
16	20.32	Chrysoeriol 7-*O*-glucoside	463.1234	−0.2	C_22_H_22_O_11_	301.0694; 286.0507	[20]
17	20.85	Malonylapiin A	651.1539	−2.6	C_29_H_30_O_17_	NA	[20]
18	21.15	Malonylapiin B	651.1547	−1.3	C_29_H_30_O_17_	519.1109; 271.0594	[20]
19	21.91	6″-Malonylapiin	651.1549	−1.0	C_29_H_30_O_17_	519.1121; 271.0556	[20]
20	22.22	Chrysoeriol 7-*O*-6″-malonyl apiosylglucoside	681.1661	−0.1	C_30_H_32_O_18_	549.1281; 301.0717; 286.0474	[20]
21	25.19	Luteolin	287.0552	0.6	C_15_H_10_O_6_	NA	[23] *
22	26.91	Sedanolide	195.1380	0.2	C_12_H_18_O_2_	177.1276; 149.1323; 125.0617; 111.0436; 97.0645; 79.0542	[23,24] *
23	27.22	Isomer of procyanidin B1	579.1487	−1.7	C_30_H_26_O_12_	427.0852; 409.0739; 291.0699; 247.0802; 205.0693	[25]
24	27.50	Apigenin	271.0605	1.5	C_15_H_10_O_5_	271.0584; 163.0352; 153.0182; 119.0480	[26] *
25	27.75	Chrysoeriol	301.0707	−0.9	C_16_H_12_O_6_	301.0710; 286.0476; 258.0500; 229.0465	[20,26]
26	27.82	Dihydroxy-3-butylphthalide	223.0969	1.9	C_12_H_14_O_4_	177.0905; 167.0327; 149.0218; 121.0271	[27]
27	28.22	Lunularic acid	257.0841	8.4	C_15_H_14_O_4_	213.0919; 171.0809; 107.0497; 106.0415	*
28	28.53	3-Butylphthalide	191.1067	0.2	C_12_H_14_O_2_	129.0684; 117.0703; 91.0526; 77.0383	[21,28]
29	28.92	Senkyunolide F	207.1016	0.1	C_12_H_14_O_3_	189.0882; 161.0959; 151.0386; 146.0726; 105.0328; 77.0381	[11,19]
30	29.56	Cinnamaldehyde	133.0648	0.1	C_9_H_8_O	133.0644; 105.0700; 79.0536; 77.0739	*
31	30.19	Sedenenolide	193.1222	−0.6	C_12_H_16_O_2_	175.1126; 147.1175; 137.0603; 105.0696; 91.0540; 77.0383	[29]
32	31.91	Linolenic acid	279.2317	−0.6	C_18_H_30_O_2_	131.0828; 95.0853; 67.0531	*

Notes: * Compound with standard reference. NA refers to the abbreviation of “Not Available” and indicates that no MS2 fragment ion was produced for this compound.

## Data Availability

All the data are included in the paper.

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
