# Peer review of "Identification of Xanthine Oxidase Inhibitors from Celery Seeds Using Affinity Ultrafiltration–Liquid Chromatography–Mass Spectrometry"

_molecules, 2023, doi:10.3390/molecules28166048_

Round 1

Reviewer 1 Report

This well-written and organised manuscript identifies xanthine oxidase inhibitors from celery seeds. After reading the paper carefully, I have only two comments:

Line 230. The boiling temperature of 70% ethanol is 78°C, not 100°C as indicated by the authors. This data should be revised.

Lines 247-249. The authors indicate that both A3 and A4 are the absorbances of the production with PBS. They should revise their description of A3 and A4.

Author Response

Response to Reviewer 1 Comments

Thanks for your attention to our previous submitted manuscript, “Manuscript ID: molecules- 2434927”. Here within enclosure is our revised manuscript, which has been improved according to the comments of the reviewers. We have seriously considered the comments and have revised the manuscript correspondingly. All the changes were tracked in red in the revised manuscript. The responses to the comments are listed one by one as follows and underline the modifications to be found easily.

Point 1: Line 230. The boiling temperature of 70% ethanol is 78°C, not 100°C as indicated by the authors. This data should be revised.

.

Response 1: Thank you for your professional advice and guidance in detail. The extraction method of celery seeds is water bath reflux extraction. After checking the experimental records, we have set the temperature of the water bath pot to 85 ℃. Under this condition, the mixture can be brought to a boil, and maintained the boil with reflux. The sentence was modified to “Dried celery seeds were extracted twice at 85 ℃ in a water bath, the first time with 8 times the amount of 70% (v/v) ethanol for 1.5 h, and the second time with 6 times the amount of 70% (v/v) ethanol for 1 h.” in line 277-279 of revised manuscript.

Point 2: Lines 247-249.The authors indicate that both A3 and A4 are the absorbances of the production with PBS. They should revise their description of A3 and A4.

Response 2: Thanks for your reminder. I am very sorry for the trouble caused to your review. The clerical error has been modified to “A3 was the absorbance of the production with PBS and XOD” in line 303-304, marked in red.

Thanks a lot for all the comments.

Kind Regards.

Reviewer 2 Report

Dear Authors,

 Your work is of interest, although you only try in vitro test , but it lacks novelty.

In this work; DOI 10.1039/d2fo01715f. “Extraction optimization, structural characterization and potential alleviation of hyperuricemia by flavone glycosides from celery seeds”. The authors already prove the inhibitory activity of apigenin and celery seeds extracts over xanthine oxidase, they also conclude, “celery seeds have a unique medicinal value in treating hyperuricemia and that the flavone glycoside extracts from celery seeds can be developed as medicine for hyperuricemia.”

 In this work, https://doi.org/10.1016/j.biopha.2023.114379, the effect of Luteolin as inhibitor of xanthine oxidase is reported.

 In my opinion, the phytochemical composition of celery seeds is well known, there is no need to publish it again, the main compounds that you found Luteolin and Apigenin have already reports over the inhibition of XO and effects on mammals, also several methods to detect Xanthine Oxidase Inhibitors are reported in the literature.

no further comments

Author Response

Response to Reviewer 2 Comments

Thanks for your attention to our previous submitted manuscript, “Manuscript ID: molecules- 2434927”. Here within enclosure is our revised manuscript, which has been improved according to the comments of the reviewers. We have seriously considered the comments and have revised the manuscript correspondingly. All the changes were tracked in red in the revised manuscript. The responses to the comments are listed one by one as follows and underline the modifications to be found easily.

Point 1: In this work; DOI 10.1039/d2fo01715f. “Extraction optimization, structural characterization and potential alleviation of hyperuricemia by flavone glycosides from celery seeds”. The authors already prove the inhibitory activity of apigenin and celery seeds extracts over xanthine oxidase, they also conclude, “celery seeds have a unique medicinal value in treating hyperuricemia and that the flavone glycoside extracts from celery seeds can be developed as medicine for hyperuricemia.” In this work, https://doi.org/10.1016/j.biopha.2023.114379, the effect of Luteolin as inhibitor of xanthine oxidase is reported.

In my opinion, the phytochemical composition of celery seeds is well known, there is no need to publish it again, the main compounds that you found Luteolin and Apigenin have already reports over the inhibition of XO and effects on mammals, also several methods to detect Xanthine Oxidase Inhibitors are reported in the literature.

Response 1: Thank you for your professional review. The articles you list above are really interesting. In the article "Extraction optimization, structural characterization and potential absorption of hyperuricemia by flavone glycosides from cell seeds", one flavone aglycone (Apigenin) and two aglycones (graveobioside A and graveobioside B) in celery seeds were separated and purified from celery seeds by traditional methods, the activity of inhibiting Xanthine oxidase and reducing uric acid was verified, which provided a good scientific basis for the potential of celery seeds in reducing uric acid.

And in https://doi.org/10.1016/j.biopha.2023.114379, by establishing a method based on surface plasmon resonance (SPR) biosensor technology, 14 Xanthine oxidase inhibitors in Chrysanthemum morifolium were screened, providing a new idea for screening Xanthine oxidase inhibitors.

The traditional steps for screening active ingredients in plants include extraction, separation purification, identification, and activity testing, which is time-consuming and labor-intensive. This study combines biological affinity with advanced analytical equipments to screen Xanthine oxidase inhibitors in celery seeds efficiently. The method we uesd effectively utilizes the advantages of high sensitivity and good selectivity of mass spectrometry, breaking through the shortcomings of traditional methods that are time-consuming and prone to loss of trace components. Through our work, seven potential xanthine oxidase inhibitors were identified, including luteolin, apigenin, chrysoeriol and four other flavonoid glycosides with Luteolin as aglycone, and one compound was identified as a new XOD inhibitor for the first time. Coincidentally, Luteolin is also an active compound in Chrysanthemum morifolium.

There are two reasons for revealing the physical composition of celery seeds. Firstly, this part of the work is a prerequisite for our subsequent screening work. Secondly, there have been numerous articles reporting on the chemical composition of celery seeds, but few have systematically elucidated the characteristic components of celery seeds by UPLC-Q/TOF as this article.

Thanks a lot for all the comments.

Kind Regards.

Reviewer 3 Report

Abstract.

IC50=1.98 mg/mL is such a high concentration when concentrations should be in terms of micrograms. Clarify.

Modify headings according to the guide for authors. Revise the entire manuscript.

Lines 36 and 37. Convert the units "6.8 mg/dL" to international units, microliter, milliliters, liters, etc., but not to deciliters (dL).

Line 231. "and then in 300 mL of fresh solution for a further 1 h" is confusing. Does it mean that the extraction process was repeated 2 times in ethanol? Clarify.

The resolution in Figure 1 should be modified. Also, the authors cut off the y-axis. Please add the complete chromatograms without cutting.

Section 2.1 is not discussed, please do it.

The authors clearly describe their results in most sections; however, the discussion is very poor, and, in some section, it is lacking. I suggest that the authors should deepen the discussion of their results and not just describe them.

The manuscript cannot be accepted in this form.

Language needs to be improved

Author Response

Response to Reviewer 3 Comments

Thanks for your attention to our previous submitted manuscript, “Manuscript ID: molecules- 2434927”. Here within enclosure is our revised manuscript, which has been improved according to the comments of the reviewers. We have seriously considered the comments and have revised the manuscript correspondingly. All the changes were tracked in red in the revised manuscript. The responses to the comments are listed one by one as follows and underline the modifications to be found easily.

Point 1: IC50=1.98 mg/mL is such a high concentration when concentrations should be in terms of micrograms. Clarify.

Response 1: Thank you for your opinion. Celery seed extract is a crude extract of dry celery seeds, and its anti-gout effect received increasing attention due to its role in XOD inhibition and sUA lowering. As a plant extract, its IC50 is indeed relatively high. However, through affinity UF-LC-MS method, some compounds with good activity were found, such as luteolin (IC50=69.23 μmol/L or 19.82 μg/mL), apigenin (IC50=92.56 μmol/L or 25.01 μg/mL), and chrysoeriol (IC50=40.52 μmol/L or 12.17 μg/mL). The IC50 of these compounds is much lower than that of celery seed extract. These studies provide a good scientific basis for explaining the uric acid lowering activity of celery seed extract, and also provide a good material basis for the subsequent screening of more Xanthine oxidase inhibitors.

Point 2:  Modify headings according to the guide for authors. Revise the entire manuscript.

Response 2: Thank a lot for your reminder. We modified all the headings according to the guide for authors.

Point 3: Lines 36 and 37. Convert the units "6.8 mg/dL" to international units, microliter, milliliters, liters, etc., but not to deciliters (dL).

Response 3: Thank you for your reminder. The sentence” The cumulative incidence of gout attacks is positively correlated with serum uric acid (sUA) levels in patients. Beyond the saturation point of MSU (6.8 mg/dL), new MSU crystals can form in joints. In contrast, a low sUA (<6 mg/dL) leads to the dissolution of existing crystals, reducing the risk of gout recurrence” in lines 34-38 was modified to “The cumulative incidence of gout attacks is positively correlated with serum uric acid (sUA) levels in patients. Beyond the saturation point of MSU (68 mg/L), new MSU crystals can form in joints. In contrast, a low sUA (<60 mg/L) leads to the dissolution of existing crystals, reducing the risk of gout recurrence” according to your suggestion.

Point 4: Line 231. "and then in 300 mL of fresh solution for a further 1 h" is confusing. Does it mean that the extraction process was repeated 2 times in ethanol? Clarify.

Response 4: Your question is very good. We apologize for any misunderstanding caused by our lack of clarity. To avoid misunderstandings, the sentence in lines 277-279 be revised as follow, “Dried celery seeds were extracted twice at 85 ℃ in a water bath, the first time with 8 times the amount of 70% (v/v) ethanol for 1.5 h, and the second time with 6 times the amount of 70% (v/v) ethanol for 1 h.”

Point 5: The resolution in Figure 1 should be modified. Also, the authors cut off the y-axis. Please add the complete chromatograms without cutting.

Response 5: Thanks for your suggestion. The resolution and format of Figure 1 have been modified as recommended.

 Point 6: Section 2.1 is not discussed, please do it.

Response 6: Thanks for your suggestion. We have added corresponding discussions on the identification results of celery seeds and the mass spectrometry fragmentation patterns of identified compounds. Please see 3.1 for details.

 Point 7: The authors clearly describe their results in most sections; however, the discussion is very poor, and, in some section, it is lacking. I suggest that the authors should deepen the discussion of their results and not just describe them.

Response 7: Thank you for your suggestion. First, we separated the results and discussion into two parts according to the requirements of the journal, and then added discussion on the identification process and Macromolecular docking results. Please see 3.1 and 3.4 for details.

Point 8: Language needs to be improved

Response 8: Thank you for your suggestion. We have reread the entire text and made language modifications to the article. Please refer to the text highlighted in red in the revised manuscript for details.

Thanks a lot for all the comments.

Kind Regards.

Round 2

Reviewer 2 Report

here is the main concern;

Mainly I recommend a major revision, the manuscript describes the chemical composition of celery and associate it  with its hypouricemic effect through their  inhibition of Xanthine oxidase (XOD), both the composition and the activity are well described in the international literature, as far as I understood the authors answer  the novelty is the isolation method of the compounds which can be useful in further studies, this fact must be clearly statement in the objectives, also the advantage of the here proposed method over the “traditional one”  must be discussed and properly justified.

Author Response

Thank you for your letter and comments concerning our manuscript, “Manuscript ID: molecules- 2434927”. Those comments are all valuable and very helpful for revising and improving our paper, as well as the important guiding significance to our research. We have seriously considered the comments and have revised the manuscript correspondingly. All the changes were tracked in red in the revised manuscript. The following is the author's response to Reviewer 2.

Point 1: Mainly I recommend a major revision, the manuscript describes the chemical composition of celery and associate it  with its hypouricemic effect through their  inhibition of Xanthine oxidase (XOD), both the composition and the activity are well described in the international literature, as far as I understood the authors answer the novelty is the isolation method of the compounds which can be useful in further studies, this fact must be clearly statement in the objectives, also the advantage of the here proposed method over the “traditional one”  must be discussed and properly justified.

.

Response 1:

Thank you for your feedback. We realize that the advantages of this study compared to traditional methods may not have been clearly expressed in previous manuscripts. We have made improvements to the original manuscript, especially in the abstract and discussion (section 3.5).

In the abstract, we add the research purpose: we aimed to establish a method based on affinity ultrafiltration-liquid chromatography-mass spectrometry (UF-LC-MS) to directly and rapidly identify the bioactive compounds contributing to the XOD inhibitory effects of celery seed crude extracts.

In the discussion, to better explain the advantages of this study compared with traditional methods and similar studies, relevant discussions are added and references recommended by the reviewer are cited. For the modification, see Section 3.5 in red, which is stated as follows.

The traditional method for screening Xanthine oxidase inhibitors from plant extracts involves multiple laborious steps, including extraction, purification, and determination of the compound library and activity detection [33, 34]. Compared to traditional procedures, UF-LC/MS combines affinity capture of active molecules against protein targets of interest with ultrafiltration and liquid chromatography‐high resolution mass spectrometry, which can simultaneously screen and identify bioactive components from complex mixtures, greatly improving efficiency, and reducing sample and reagent consumption [35, 36].

Many previous affinity UF-LC-MS studies used mass spectrometry to identify the compounds in the extracts and determined the candidates by comparing the peak area of each compound in the liquid chromatogram [37, 38]. However, co-elution and peak overlapping often occurred in the liquid chromatogram [39], and some compounds, such as saponins, alkaloids and amines, have poor UV absorption, resulting in missed screening. The extracted ion current chromatogram used in this study selects specific ion fragments for peak area comparison, effectively avoiding the pitfalls of conventional methods and increasing the hitting rate.

In addition, the authors carefully checked the references of the full text to ensure that all references are relevant to the content of the article.

All the authors are grateful for the reviewer's comments and corrections.

Kind Regards.

Xiaodong Liu, PhD,in Aug 2023

Department of Anaesthesia and Intensive Care,

The Chinese University of Hong Kong.

Tel: (852)3505 2740;

Reviewer 3 Report

I revised the manuscript and the authors addressed most of the comments and suggestions. In my opinion the manuscript has been improved and can now be accepted for publication.

Author Response

Thanks a lot for your nice feedback. Your previous comments are all valuable and very helpful for revising and improving our paper, as well as the important guiding significance to our research.

Round 3

Reviewer 2 Report

No further comments, in my opinion the work is ready for publication

No further comments.